# PEGylated Magnetic Nano-Assemblies as Contrast Agents for Effective T2-Weighted MR Imaging

**DOI:** 10.3390/nano9030410

**Published:** 2019-03-11

**Authors:** Byunghoon Kang, Jaewoo Lim, Hye-young Son, Yuna Choi, Taejoon Kang, Juyeon Jung, Yong-Min Huh, Seungjoo Haam, Eun-Kyung Lim

**Affiliations:** 1Department of Chemical and Biomolecular Engineering, Yonsei University, Seoul 03722, Korea; vv345vv345@naver.com; 2BioNanotechnology Research Center, Korea Research Institute of Bioscience and Biotechnology (KRIBB), 125 Gwahak-ro, Yuseong-gu, Daejeon 34141, Korea; zeuyim5052@kribb.re.kr (J.L.); kangtaejoon@kribb.re.kr (T.K.); jjung@kribb.re.kr (J.J.); 3Department of Nanobiotechnology, KRIBB School of Biotechnology, University of Science & Technology (UST), 125 Gwahak-ro, Yuseong-gu, Daejeon 34113, Korea; 4Department of Radiology, College of Medicine, Yonsei University, Seoul 03722, Korea; shy916@yuhs.ac (H.-Y.S.); YUNA517@yuhs.ac (Y.C.); 5Severance Biomedical Science Institute, College of Medicine, Yonsei University, Seoul 03722, Korea; 6YUHS-KRIBB Medical Convergence Research Institute, Seoul 03722, Korea

**Keywords:** Magnetic resonance image, PEGylated, Poly(ethylene glycol)-poly(lactic acid), contrast agent

## Abstract

We designed a high-sensitivity magnetic resonance imaging contrast agent that could be used to diagnose diseases. First, magnetic nanocrystals were synthesized by a thermal decomposition method on an organic solvent to obtain a high magnetism and methoxy poly(ethylene glycol)-poly(lactic acid) as an amphiphilic polymer using the ring-opening polymerization method to stably disperse the magnetic nanocrystals in an aqueous phase. Subsequently, the magnetic nanoclusters simultaneously self-assembled with methoxy poly(ethylene glycol)-poly(lactic acid) using the nano-emulsion method to form magnetic nanoclusters. Because their shape was similar to a raspberry, they were named PEGylated magnetic nano-assemblies. The PEGylated magnetic nano-assemblies were dispersed stably in the aqueous phase with a uniform size of approximately 65–70 nm for an extended period (0 days: 68.8 ± 5.1 nm, 33 days: 69.2 ± 2.0 nm, and 44 days: 63.2 ± 5.6). They exhibited both enough of a magnetic resonance (MR) contrast effect and biocompatibility. In an in vivo study, the PEGylated magnetic nano-assemblies provided a high contrast effect for magnetic resonance images for a long time after one treatment, thereby improving the diagnostic visibility of the disease site.

## 1. Introduction

Magnetic resonance (MR) imaging is the best investigative tool to obtain tomographic images with a high resolution and to offer excellent anatomical information in living organisms. Its detection capability can be greatly enhanced using MR contrast agents that enable noninvasive monitoring and disease identification [1,2,3,4,5,6,7,8,9,10,11,12,13,14,15,16,17]. Magnetic nanocrystals synthesized in an organic phase have well-defined crystallized structures and a high magnetic sensitivity, and they have been reported to be used as contrast agents [4,18,19]. Of course, since they are dispersed in the organic phase, additional surface modification with a hydrophilic layer (e.g., polyethylene glycol (PEG)) is required to improve their colloidal stability in the aqueous phase [7,9,10,12,20,21,22]. PEG has a high hydrophilicity, a low cytotoxicity, as well as a high cell permeability; in particular, it contributes to diminishing nonspecific interactions with serum proteins by forming hydrogen bonds with water molecules, leading to an increasing residence (circulation) time in the blood [12,15,23,24,25,26,27,28,29,30,31,32,33,34,35,36,37,38,39]. Two main surface modification methods are used for PEG-based materials: (1) the exchange method, in which the hydrophobic ligand on the magnetic nanocrystal (MNC) surface is exchanged with PEG at a high temperature, and (2) the addition method, in which the surface of a hydrophobic MNC is wrapped (or coated or covered) with the emulsion method using a PEG-based amphiphilic polymer as a surfactant [23,24,25,27,31,32,33,34,35,36,37,38,40,41,42]. This addition method enables the fabrication of a multifunctional nanocomposite by simultaneously loading drugs and fluorescent materials as well as MNCs. However, obtaining uniformly sized nanoparticles, especially nanoclusters, is not easy. There is a limitation in increasing the size of MNCs to enhance the saturation magnetization (Ms) because it also induces the transition between the supermagnetic-ferrimagnetic transitions. Instead, it has been reported that a method of maintaining the superparamagnetic behaviour with a high magnetization by forming magnetic nanoclusters (or assemblies) is effective. Magnetic nanoparticles composed of multiple single MNCs are particularly attractive due to their high magnetic susceptibility, low coercive force, and high magnetic properties [4,43,44].

Herein, we have developed high-sensitivity MRI contrast agents that improve the diagnostic visibility of the disease site. We synthesized mPEG-PLA as a surfactant and then wrapped (encapsulated) hydrophobic 12-nm MNCs with mPEG-PLA using the nano-emulsion method. The particles are formed by homogeneously aggregating MNCs that are dispersed stably in a water phase and are named PEGylate magnetic nano-assemblies (PEGylated MNs). 

## 2. Materials and Methods 

### 2.1. Materials 

Methoxy polyethylene glycol (mPEG) (5 kDa), stannous octoate (Sn(Oct)_2_), 3,6-dimethyl-1,4-dioxane-2,5-dione (lactide; LA), chloroform, iron (III) acetylacteonate (Fe(acac)_3_), manganese (II) acetylacetonate (Mn(acac)_2_), 1,2-hexadexanediol, lauric acid, laurylamine, and benzyl ether were purchased from Sigma-Aldrich (St. Louis, MO, USA). All other chemicals and reagents were of analytical grade. The cell proliferation kit I (MTT) was purchased from Roche (Basel, Switzerland)_._


### 2.2. Synthesis of Magnetic Nanocrystals (MNCs)

As previously described, we synthesized hydrophobic magnetic nanocrystals (MNCs and MnFe_2_O_4_) in an organic solvent using the thermal decomposition method [4,7,9,10,12,18,19,20,21,22]; 2 mmol of Fe(acac)_3_, 1 mmol of Mn(acac)_2_, 10 mmol of 1,2-hexadecanediol, 6 mmol of lauric acid, and 6 mmol of laurylamine were dissolved in benzyl ether (20 mL) under an ambient nitrogen atmosphere. This mixture was preheated to 200 °C for 2 h under uniform stirring and then refluxed at 300 °C for 1 h. After completion of this reaction, the reactants were cooled down to room temperature and purified with an excess of ethyl alcohol (99.9%) for the products. The MNCs of about 12 nm were obtained using the seed-mediated growth method. 

### 2.3. Synthesis of mPEG-PLA by Ring-Opening Polymerization

The methoxy poly(ethylene glycol)-poly(lactic acid) (mPEG-PLA) as amphiphilic polymers were synthesized by the ring opening polymerization (ROP) method, which the polymerized lactide monomers in the presence of mPEG and Sn(Oct)_2_ were the catalyst through the use of the bulk method. Briefly, dried mPEG (2 g, 0.4 μmol), lactide (0.8 g, 5.55 μmol), and Sn(Oct)_2_ (100 μL) were completely dissolved in 50 mL of toluene at 120 °C under a nitrogen atmosphere. After completely melting for 2 h, the mixture was slowly heated to 180 °C for 4 h. After the reaction was terminated, the heat source was removed from the reactant and the temperature was cooled down to room temperature. The solvent of this reactant was rapidly eliminated using a rotary evaporator (50 HZ, EYELA). For purification, this reactant was redissolved in toluene (2 mL) and precipitated in excess cold diethyl ether, and then, it was filtered using a vacuum filtration. This purification process was repeated three times. The reactant was freeze-dried to obtain a purified product as a white powder and stored under a vacuum before use. The synthesized mPEG-PLA was analyzed using FTIR and ^1^H-NMR spectroscopy with CDCl_3_. The critical micelle concentration (CMC) of mPEG-PLA was measured by conductivity. The conductivity (μS cm^−1^) of the mPEG-PLA solutions with various concentrations was measured using a conductivity meter. The mPEG-PLA concentration was plotted against the conductivity. The CMC was identified as the point on a plot where the slope changes. 

### 2.4. Preparation of PEGylated Magnetic Nano-Assemblies (PEGylated MNs)

The PEGylated MNs were prepared by a nano-emulsion method. The mPEG-PLA solution and MNC-containing organic phase were prepared by dissolving mPEG-PLA (100 mg) and MNC (20 mg) in deionized water (20 mL) and methylene chloride (4 mL), respectively.

Afterwards, the MNC-containing organic solvent was poured into the mPEG-PLA solution. This solution was ultra-sonicated in an ice bath for 20 min at 450 rpm and stirred overnight at room temperature to evaporate methylene chloride. The resulting suspension was centrifuged 3 times for 30 min each at 18,000 rpm. After the supernatant was removed, the precipitated PEGylated MNs was re-dispersed in 5 mL of DI water. After the preparation, the size distributions and surface charges of the PEGylated MNVs were analyzed by a particle size and zeta-potential analyzer (ELS-Z, Otsuka Electronics). Additionally, its colloidal stability was analyzed by measuring the size variation, monitoring for 44 days. Their morphologies and magnetic properties were analyzed by transmittance electron microscopy (TEM, TECNAI G2, FEI) and a vibrating-sample magnetometer (VSM) at 298K, respectively. The weight ratio (%) of mPEG-PLA in the samples was measured using thermos-gravimetric analysis. Finally, the relaxivity (R2) data were obtained by using magnetic resonance (MR) imaging analysis. 

### 2.5. Biocompatibility Tests

The biocompatibility of mPEG-PLA and PEGylated MNs against NIH3T6.7 cells was evaluated by measuring the inhibition of cell proliferation using the MTT assay kit. The NIH3T6.7 cells were maintained in Dulbecco’s modified eagle′s medium (DMEM) containing fetal bovine serum (FBS) (10%) and antibiotics (1%) at 37 °C in a humidified atmosphere with 5% CO_2_. Seeded were 10^4^ cells per well into a 96-well plate, and they were incubated at 37 °C overnight to attach onto the wells. Then, they were treated with various concentrations of mPEG-PLA and the PEGylated MN solution and further incubated for 24 h. The MTT assay was performed depending on the procedure recommended by the manual. The cell viability (%) was determined as the ratio of purple intensity in viable cells treated with mPEG-PLA and PEGylated MNs to the intensity in nontreated (control) cells. The cell viability was normalized to that of the control cells (which were considered to have 100% cell viability).

### 2.6. In Vivo (Mouse Xenograft Tumor) Model Procedure

We performed mouse xenograft in tumor models, in which NIH3T6.7 cells (10^7^ cells in 50 μL of saline per animal) were implanted in the proximal thigh region of BALB/c-nude mice (4–5 weeks of age) [7,11,45]. Then, the MR imaging was performed using 5 mice 4 weeks after the tumor cell transplantation. All animal experiments were conducted with the approval of the Association for Assessment and Accreditation of Laboratory Animal Care (AAALAC) International. 

### 2.7. MR Imaging Procedure

We performed the MR imaging experiment of a PEGylated MN solution with a 1.5 T clinical MRI instrument with a micro47 surface coil (Intera; Philips Medical Systems, Best, the Netherlands). The T2 weights of various concentrations of PEGylated MN solution were measured by the Carr–Purcell–Meiboom–Gill (CPMG) sequence at room temperature with the following parameters: TR = 10 s, 32 echoes with a 12-ms even echo space, number of acquisitions = 1, point resolution of 156 × 156 mm, and section thickness of 0.6 mm. The relaxivity coefficient (mM^−1^ s^−1^) was equal to the ratio of R2 (1/T2, S^−1^) to the PEGylated MN concentration. In addition, in vivo MR imaging experiments were performed with a 3 T clinical MRI instrument with a micro-47 surface coil (Intera; Philips Medical Systems, Best, the Netherlands). The T2 weights of a nude mouse injected with PEGylated MNs were measured by the CPMG sequence at room temperature with the following parameters: TR = 10 s, 32 echoes with a 12-ms even echo space, number of acquisitions = 1, point resolution of 156 × 156 mm, and section thickness of 0.6 mm. For the T2-weighted MR imaging of the nude mouse model, we adopted the following parameters: resolution of 234 × 234 mm, section thickness of 2.0 mm, TE = 60 ms, TR = 4000 ms, and number of acquisitions = 1.

## 3. Results and Discussion

### 3.1. Synthesis and Characterization of mPEG-PLA 

We synthesized mPEG-PLA as an amphiphilic copolymer that could effectively encapsulate hydrophobic magnetic nanocrystals through ring-opening polymerization (Figure 1) [23,24,25,27,31,32,33,34,35,36,37,38,41]. mPEG(5K)-PLA(2K) was obtained by controlling the feed ratio (LA/EG), and its chemical structure was confirmed using FTIR and an ^1^H-NMR spectrometer (Figure 2) [28,33,35,36,37,43]. The hydroxyl groups at the end of mPEG acted as the initiators, and the lactide monomers for the attachment and growth of the PLA blocks were located at the end of PEG. mPEG-PLA commonly showed an absorption that was assigned to the C–O–C band of mPEG (1087–1184 cm^−1^), the –C=O stretching of lactide (LA) (1750–1760 cm^−1^), and the –CH_2_– stretching of mPEG (2850–2950 cm^−1^). However, compared with that of LA, the strong peak at 1150–1300 cm^−1^ indicated the C–C(=O)–O vibration of LA disappeared (Figure 2a). As shown in Figure 2b, the peaks corresponding to the methylene (–CH_2_–) and methoxy end (CH_3_O–) groups of mPEG were still confirmed at approximately 3.38 ppm and 3.64 ppm, respectively, but the peak corresponding to the hydroxyl (–OH) group was no longer observed at 4.79 ppm. Additionally, the methyne (CH) and methyl (–CH_3_) groups were observed in PLA at 5.17 ppm and 1.60 ppm, respectively. These spectra indicated that mPEG-PLA was successfully synthesized. We also measured the conductivity of mPEG-PLA to confirm its applicability as an amphiphilic polymer. The conductivity increases almost linearly with the mPEG-PLA concentration; after a certain concentration is reached, the conductivity increases linearly with the lower slope. This concentration at which the slope of conductivity changes was identified as the critical micelle concentration (CMC) (Figure 3) [39,46]. This result indicated that the surfactant adsorption at the interface did not increase until the mPEG-PLA concentration reached 5.496 μM; however, the further addition of mPEG-PLA resulted in the formation of micelles in the solution.

### 3.2. Formulation and Characterization of PEGylated MNs 

We fabricated PEGylated magnetic nanoparticles composed of hydrophobic magnetic nanocrystals (MNCs) encapsulated by mPEG-PLA using the nano-emulsion method [3,7,8,9,10,11,12,13,16,20,21,22,39,45,47,48]. The morphologies of these particles confirmed by TEM showed that MNCs were clustered spherically (Figure 4a). Based on this TEM image, we denoted this nanoparticle as a PEGylated magnetic nanoberry (PEGylated MN). We also observed that their stability changed over time and measured their sizes and surface charges using laser scattering (0 day: 68.8 ± 5.1 nm and 0.7 ± 0.3 mV, 33 day: 69.2 ± 2.0 nm and −2.4 ± 0.7 mV, and 44 day: 63.2 ± 5.6 nm and −1.1 ± 0.9 mV, respectively). After PEGylated MN fabrication, we visually observed their colloidal stability for 15 day. The day when the particles were produced was referred to as day 0. As shown in Figure 4b, these PEGylated MNs did not agglomerate for a long time (15 days) and remained very stable in the aqueous phase. Additionally, their sizes and charges barely changed over a 44-day period due to the PEGylation effect [23,26,28,29,30]. This finding signifies that the surface of the PEGylated MNs was well-coated with mPEG-PLA. Therefore, these particles could be stably presented in the water phase because of the hydrogen bonding with PEG molecules on the surface of PEGylated MNs and water molecules. As well, the poly dispersity index (PDI) values were calculated based on the dynamic light scattering (DLS) analysis results. When each value was substituted into the PDI equation (= Standard deviation^2^/particle size), the PDI values were 0.38, 0.06, and 0.49, respectively (Table 1 and Appendix A). The PDI values of the particles ranged from 0 to 0.0.8 in the nearly monodispersed and 0.08 to 0.7 in the uniformly dispersed [49]. Based on our PDI values, we judged that PEGylated MN exhibited uniform dispersion and was acceptable to use in the pharmaceutical filed. 

The weight (%) of the MNCs in PEGylated MNs was analyzed by a thermogravimetric analysis as the obtained amount of MNCs divided by the initial amount of PEGylated MNs. It was confirmed that the magnetic sensitivity of MNCs was affected by their size and composition [44,46]. Of the 4 types pf MNCs (MnFe_2_O_4_, Fe_3_O_4_, CoFe_2_O_4_, and NiFe_2_O_4_), MnFe_2_O4 nanocrystals showed the strongest MR contrast effect, with high relaxivity values. Especially, 12-nm MnFe_2_O_4_ nanocrystals exhibited the highest mass magnetization value. Based on this previous references, we used 12-nm MnFe_2_O_4_ nanocrystals as MNCs to fabricate PEGylated MNs. A total of 81.3 wt% of MNCs was loaded into the PEGylated MNs (Figure 5a). Despite the 18.7 wt% mPEG-PLA coating, the superparamagnetism of 12-nm MNCs was maintained with a high saturated magnetization value (40.3 emu/g_MNCs_) at 298 K (Figure 5b) [4,43]. These results demonstrated that PEGylated MNs well-covered with mPEG-PLA exhibited high magnetic properties. 

We evaluated the feasibility of PEGylated MNs as MR imaging agents and measured their MR signal intensities in an aqueous phase under various concentrations at room temperature. In Figure 6, the T2-weighted MR image showed an increasingly stronger black as the concentration increased, and their T2 values were measured. The relaxivity (r2) of PEGylated MNs was calculated from the least-square curve fitting of the MNC concentration versus relaxation rate (R2, S^−1^). The corresponding r2 coefficient was determined to be 217.07 mM^−1^ s^−1^, that was higher (approximately 1.13-fold) than that of commercial MRI contrast media (Ferumoxide: 190.5 mM^−1^ s^−1^) [46,50]. These findings demonstrated that PEGylated MNs possessed a remarkably high MR imaging effect due to the enhanced magnetism through the dense clustering of a large amount of 12-nm MnFe_2_O_4_ (MNCs) in the PEGylated MNs [4,43,44].

### 3.3. Biocompatibility of the PEGylated MNs

Before applying the particles for in vivo imaging, first we evaluated the biocompatibility of the PEGylated MNs against NIH3T6.7 cells using a MTT assay that was performed after incubation with various concentrations overnight. The cell viability was maintained above 80% without inhibitory effects on the cell proliferation within the PEGylated MN concentration range. Even when inorganic MNCs were present in them, no effect on the cell viability was observed compared to that of mPEG-PLA (Figure 7) [23,27,31,32,33,34,35,36,37,41,42]. 

### 3.4. In Vivo MR Imaging 

Next, we performed MR imaging in mouse xenograft tumor models using PEGylated MNs to evaluate their diagnostic ability as MR imaging contrast agents. First, BALB/c-nude mice were subcutaneously implanted in the proximal thigh with NIH3T6.7 cells for the xenograft mouse model, and PEGylated MNs (200 μg_Fe+Mn_ in 200 μL buffer) were injected into the tail vein of the mouse (intravenous injection). After a PEGylated MN injection, we observed that the tumor region became gradually darker along the intra-tumoral blood vessels with a high MR signal intensity (Figure 8a). In particular, the MR signal at the tumor region was consistently enhanced with darkening on the MR image by increasing the ΔR2/R2_Pre_ value by about 16.0% at 7 h postinjection relative to the preinjection intensity (Figure 8c). Additionally, we clearly identified the MR signal changes by color-mapping the tumor areas of the MR image. The color map image was intended to make the R2 value (MR signal) change value visually clear. In a color map image, as the MR signal increases, the color changes from blue to red. Therefore, in Figure 8b, the red color spread out gradually along the vascular distribution as the MR signal increased, similar to the T2-weighted MR images. These results indicated that PEGylated MNs were capable of sufficiently long-term circulation in the bloodstream (body) due to their physical properties (e.g., PEGylation), thereby enabling the accumulation in tumor tissues by the enhanced permeability and retention (EPR) effect. 

## 4. Conclusions

In this study, we synthesized mPEG-PLA as an amphiphilic polymer using the ring-opening polymerization method and then encapsulated the hydrophobic magnetic nanocrystals (MNCs) in an organic solvent with mPEG-PLA using the nano-emulsion method to allow the stable dispersion in the aqueous phase. At this time, the MNCs were uniformly assembled in raspberry form, and we named it PEGylated magnetic nano-assemblies (PEGylated MNs). The PEGylated MNs exhibited a good stability in an aqueous phase for an extended time due to the PEG molecules on the particle surface (PEGylation effect) as well as an enough of an MR contrast effect due to the magnetic clustering effect compared to those of a commercial MR contrast agent. In addition, we confirmed that PEGylated MNs had a potential use as MRI agents for cancer detection through in vivo studies. 

## Figures and Tables

**Figure 1 nanomaterials-09-00410-f001:**
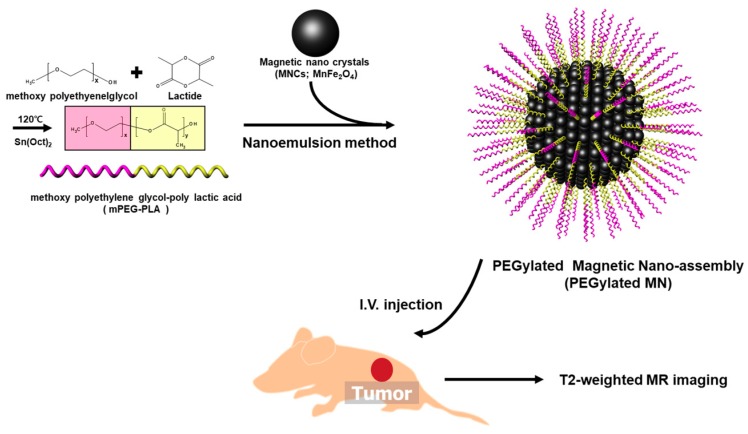
A schematic illustration of PEGylated magnetic nano-assembly (PEGylated MN) synthesis using mPEG-PLA and magnetic nanocrystals (MNC_S_) as an effective MRI contrast agent.

**Figure 2 nanomaterials-09-00410-f002:**
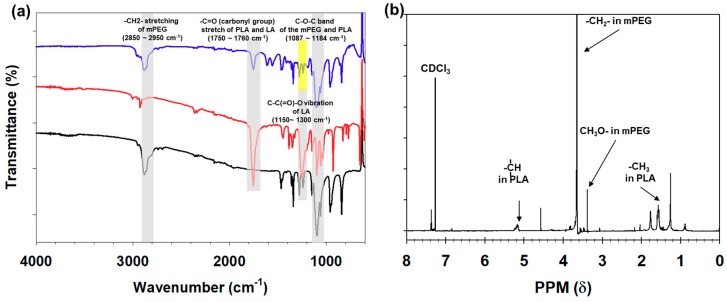
(**a**) The FTIR spectra of mPEG (5K) (black), lactide (red), and mPEG-PLA (blue) and (**b**) the ^1^H-NMR spectrum of mPEG-PLA (CDCl_3_: Solvent).

**Figure 3 nanomaterials-09-00410-f003:**
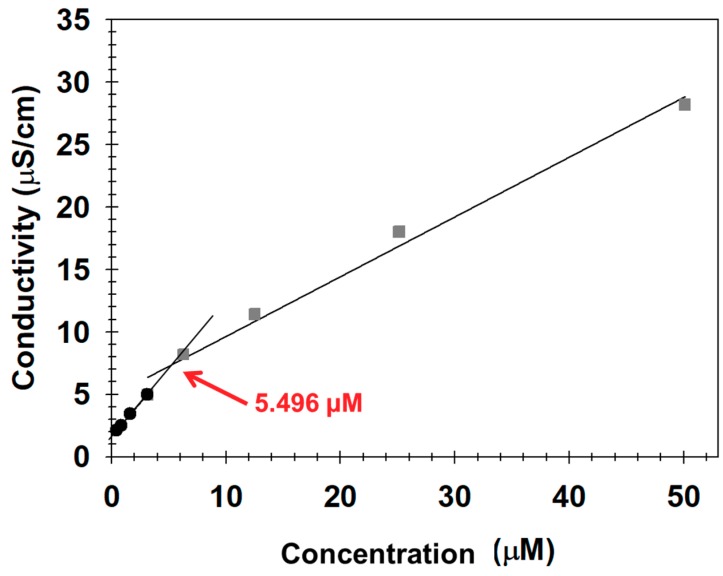
The determination of the critical micelle concentration (CMC) of mPEG-PLA using conductivity.

**Figure 4 nanomaterials-09-00410-f004:**
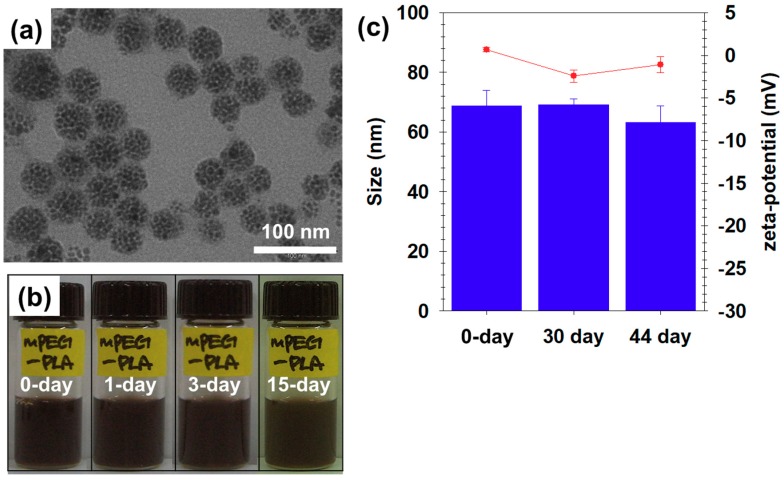
(**a**) The transmission electron microscopy (TEM) images of PEGylated MNs, (**b**) their colloidal stability for 15 days, and (**c**) their size distribution (blue bars) and zeta-potential (red circles) analyses over 44 days.

**Figure 5 nanomaterials-09-00410-f005:**
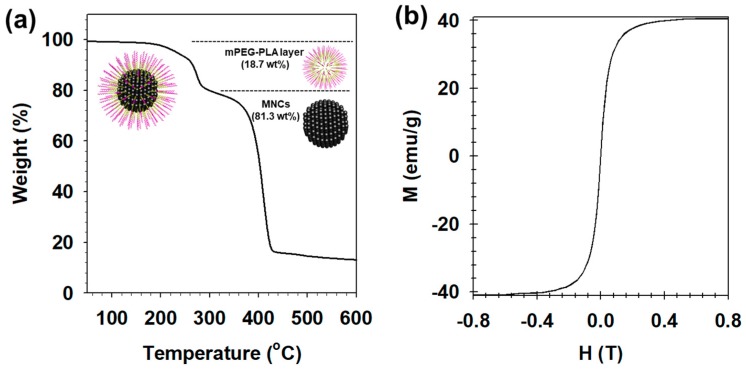
(**a**) The magnetic hysteresis loops and (**b**) thermogravimetric analysis (TGA) of the PEGylated MNs.

**Figure 6 nanomaterials-09-00410-f006:**
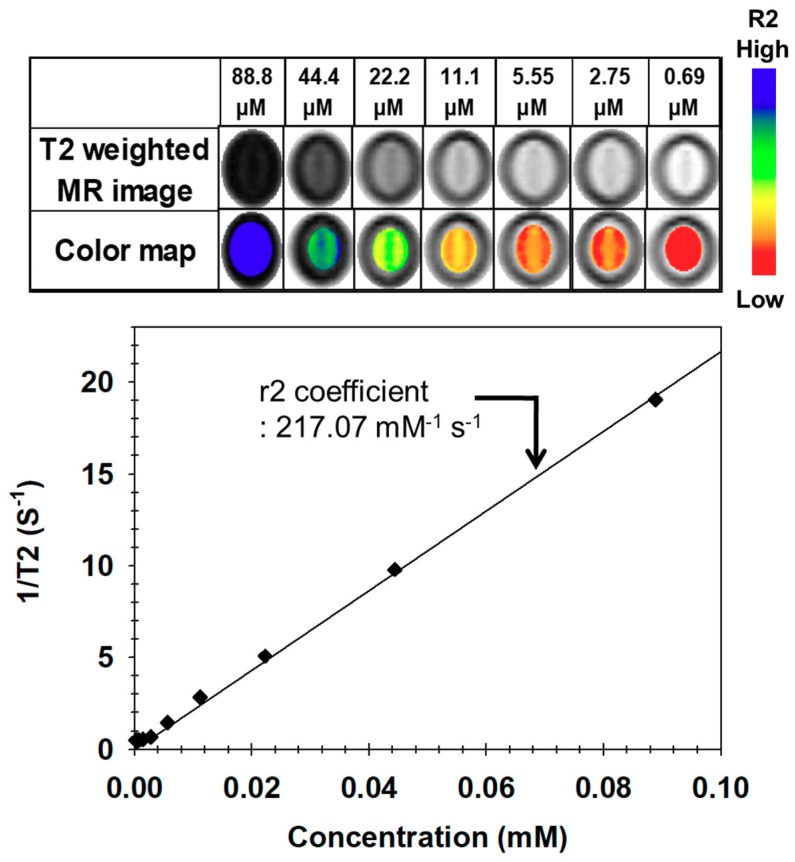
The T2-weighted MR images of a PEGylated MN solution and their color maps and their 1/T2 (S^−1^) values at 1.5 T.

**Figure 7 nanomaterials-09-00410-f007:**
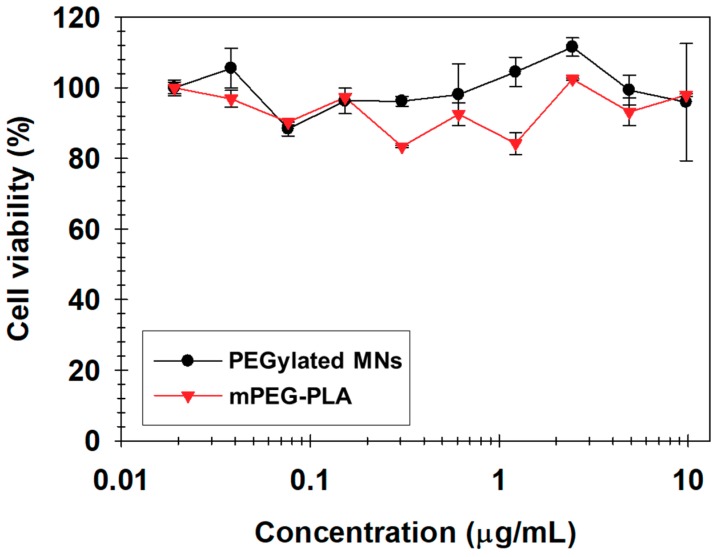
The viability of cells treated with mPEG-PLA (▼) and PEGylated MN (●) at various concentrations.

**Figure 8 nanomaterials-09-00410-f008:**
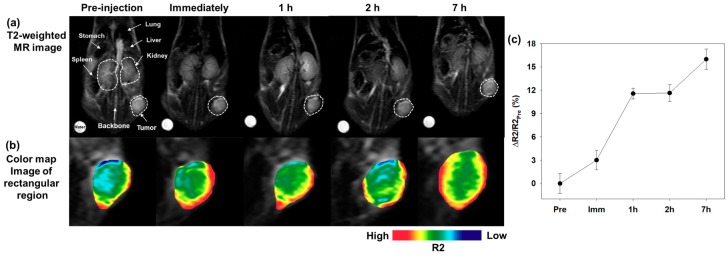
(**a**) The T2-weighted MR images of mice and (**b**) the color-map images of the polygonal region with the white dashed line of Figure 8a. (**c**) The ΔR2/R2_Pre_ (%) graph of the T2-weighted MR images versus the time after an intravenous venous (I.V.) injection of PEGylated MNs (Pre: preinjection, IMM: immediately following the injection, 1 h: 1 h following the injection, 3 h: 3 h following the injection, and 7 h: 7 h following the injection). A 3.0 T human MR scanner was used.

**Table 1 nanomaterials-09-00410-t001:** The size, poly dispersity index (PDI) values, and zeta potential data of PEGylated MN over 44 days.

Time	Size (nm)	PDI ^a^	Zeta (mV)
0 day	68.8 ± 5.1	0.38	0.7 ± 0.3
33 days	69.2 ± 2.0	0.06	−2.4 ± 0.7
44 days	63.2 ± 5.6	0.49	−1.1 ± 0.9

All data are depicted as the mean ± S.D, and N > 3, ^a^ PDI = (S.D.)^2^/Avg. size.

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
