# Peer review of "PEGylated Magnetic Nano-Assemblies as Contrast Agents for Effective T2-Weighted MR Imaging"

_nanomaterials, 2019, doi:10.3390/nano9030410_

Reviewer 1 Report

The manuscript “PEGylated magnetic nanoberries as T2-weighted MR contrast agents….” by Lim and co-workers describes the preparation, characterisation, and MR application of polymer stabilised magnetic nanoparticle assemblies using a polymer mediated clustering approach. There has been significant work in this area, but questions about how MR contrast is generated and what the realisable maximum values are remain open. The work is mostly well described and is of interest to the readership of this journal. There are some flaws which need to be addressed; a resubmission may be of publishable quality.

The abstract should be rewritten removing all acronyms, also what is magnetic sensitivity?

The polymer synthesis is incompletely described. There is no significant work-up, little purification and no demonstration that a pure compound has been prepared. In the absence of such data or a demonstration that the reaction is quantitative (and equimolar quantities were used) it is hard to know what has been prepared. This is important as any impurities may significantly affect the clustering step. Dialysis might help, but purity could be demonstrated by chromatography, MS, or perhaps 1H NMR end group analysis, and this should be included.

It would be better not to call to nanoberries nanoparticles, perhaps clusters or assemblies?

The DLS analysis is incomplete; the uncertainties given are presumably from repeat measurements. What was the PDI and how do the dhyd and PDI compare to the statistical TEM analysis? The Malvern instrument was not used, but some measure of polydispersity must be possible. The suspensions are stable in that the dhyd value didn’t change, but are the PDI and concentration unchanging (the DLS counts could help for the latter).

My major criticism is the absence of a comparative analysis of the relaxivity versus the literature; there are a great many papers including a number on polymer stabilised clusters of this type. I know some of this work and r2=217s-1mM-1 (1.5T) at c.65 nm seems reasonable if not very high, but it is for the Authors to complete this analysis. There are at least three points; the MNCs size and crystallinity; the cluster size, and; the polymer (weight%, mol wt and relative wts of the two blocks). 

Finally, there is a lot of information missing from the MRI part:

How many subjects were analysed? The absence of error bars in Fig 8 suggests only one. If that is the case it is insufficient for a figure.

Why were the other organs not analysed?

How was the sample introduced, e.g. bolus tail-vein injection? If so at what volume, and at what concentration? What was the medium isotonic?

Author Response

Reviewer 1

The manuscript “PEGylated magnetic nanoberries as T2-weighted MR contrast agents….” by Lim and co-workers describes the preparation, characterization, and MR application of polymer stabilised magnetic nanoparticle assemblies using a polymer mediated clustering approach. There has been significant work in this area, but questions about how MR contrast is generated and what the realisable maximum values are remain open. The work is mostly well described and is of interest to the readership of this journal. There are some flaws which need to be addressed; a resubmission may be of publishable quality.

1. The abstract should be rewritten removing all acronyms, also what is magnetic sensitivity?

: We have revised the abstract as a whole.

2. The polymer synthesis is incompletely described. There is no significant work-up, little purification and no demonstration that a pure compound has been prepared. In the absence of such data or a demonstration that the reaction is quantitative (and equimolar quantities were used) it is hard to know what has been prepared. This is important as any impurities may significantly affect the clustering step. Dialysis might help, but purity could be demonstrated by chromatography, MS, or perhaps 1H NMR end group analysis, and this should be included.

: The polymer synthesis section has been modified in detail.

Page 8, line 196: After the reaction was terminated, the heat source was removed from reactant and their temperature was cooled down to room temperature. The solvent of this reactant was rapidly eliminated using rotary evaporator (50 HZ, EYELA). For purification, this reactant was re-dissolved in toluene (2 mL) and precipitated in excess cold diethyl ether, and then it was filtered using a vacuum filtration. This purification process was repeated three times. The reactant was freeze-dried to obtain purified product as a white powder and stored under a vacuum before use.

3. It would be better not to call to nanoberries nanoparticles, perhaps clusters or assemblies?

: As advised by the reviewer, we have changed nanoparticle’s name to nano-assemblies. 

4. The DLS analysis is incomplete; the uncertainties given are presumably from repeat measurements. What was the PDI and how do the dhyd and PDI compare to the statistical TEM analysis? The Malvern instrument was not used, but some measure of polydispersity must be possible. The suspensions are stable in that the dhyd value didn’t change, but are the PDI and concentration unchanging (the DLS counts could help for the latter).

: As reviewer’s comment, PDI values were calculated based on the DLS analysis results.

                                               -------------------------------------------------------------- (1)

According to various literature, PDI value ranges from 0 to 1 (0 ~ 1) and indicates:

0-0.08: Nearly monodisperse sample

0.08 to 0.7: Mid-range value 

0.7-1: very broad distribution of particle sizes

Mean diameter and standard deviation of PEGylated MNs at 0 days were 68.8 nm and 5.1 nm, respectively. When each value is substituted into the above equation (1), the PDI value is about 0.378.  Based on this value, we judged that our PEGylated MNs are uniformly disperse and acceptable to use in the pharmaceutical filed.

5. My major criticism is the absence of a comparative analysis of the relaxivity versus the literature; there are a great many papers including a number on polymer stabilized clusters of this type. I know some of this work and r2=217s-1mM-1 (1.5T) at c.65 nm seems reasonable if not very high, but it is for the Authors to complete this analysis. There are at least three points; the MNCs size and crystallinity; the cluster size, and; the polymer (weight%, mol wt and relative wts of the two blocks). 

: We thank for reviewer’s valuable comment. As the reviewer mentioned, magnetic relaxivity is influences by three factors: size and crystallinity of magnetic nanocrystals (MNCs), and magnetic nanoparticles’ size. First, the size of MNCs is directly dependent on the magnetic sensitivity (relaxivity, r2) which has been experimentally proven by many researchers.1-7 Cheon’s group reported the effect of magnetic sensitivity according to the size and composition of MNCs.4-5 MnFe2O4 nanoparticle showed the strongest MR contrast effect, with high relaxivity value (Figure R1). Especially, 12-nm MnFe2O4 nanoparticles showed the highest mass magnetization value (Figure R2).5

Figure R1. Magnetism-engineered iron oxide nanoparticles and effects of their magnetic spin on MRI, which is reproduced from ref 5.

Figure R2. Size-dependent MR contrast effect of MnFe2O4 and Fe3O4 nanoparticles, which is reproduced from ref 5.

In addition, for the r2 value of magnetic nanoparticles including a single magnetic nanoparticle and a magnetic nanocluster, various theoretical and experimental studies have been also investigated so far. According to the descriptions of H. Weller et. al., the r2 values are given by three different regimes with increasing MNC diameter: i) an increase in the so-called motional average regime (MAR), ii) a maximum in the static dephasing regime (SDR), and iii) an echo-dependent decrease of r2 in the echo-limiting regime (ELR). In the MAR, the r2 for MNCs increases with increasing their size, showing a typical behavior of MNPs. In contrast, for a further increase in size of MNCs, the r2 no longer increases and subsequently decreases in the SDR and MAR conditions.6 R. Weissleder et. al. also demonstrated similar overall r2 behavior for various sized MNPs and MNCs by combining the results from the chemical exchange (CE) and static dephasing (SD) models.7 In their study, they synthesized clusters of manganese-doped ferrite MNPs (Mn-MNPs) by embedding them into a thin silica shell and compared the r2 values of single core Mn-MNPs with those of clusters as a function of particle size. Notably, in the SD regime, the r2 is independent of particle size (r2 ~ a0, where the a is the overall radius of particle) and reaches the absolute limit of r2. As the particle diameter increases beyond the SD regime, the r2 of NMPs begins to decrease (r2 ~ a-1) because the magnetic field generated by the particles diminishes due to their thick shells (Figure R3). In that respect, therefore, it is hardly seen that the r2 value of MNCs is always proportional to their size.

Figure R3. The r2 curves for Mn-doped ferrite MNPs (Mn-MNPs) and NMPs (containing a cluster of Mn-MNPs), which is reproduced from ref. 5.

Another behaviour of MNCs associated with the r2 value is that they have stronger magnetic dipole-dipole interactions than those of individual magnetic nanoparticle as a result of the distance-dependent nature of such interactions. C. Mao et. al. reported that the blocking temperature (TB) had shifted from 111 to 181 K after cluster formation, which indicates strong dipole-dipole interactions had taken place among nanoparticles inside the clusters (Figure R4).8 Thus, the higher TB of the clusters with densely packed individual nanoparticles leads to higher magnetic anisotropy (K), which is given by the following equation: K = 25κBTB / V, with κB as the Boltzmann constant and V as the volume of a nanoparticle. The enhanced K values of such clusters also bring to the r2 enhancement. This is called magnetic clustering effect.

Figure R4. ZFC-FC curves of a single and clustered MNPs measured at 50 Oe, which is reproduced from ref 8.

Therefore, there is a limitation in increasing the size of MNCs to enhance saturation magnetization (Ms) because it also induces the transition between the supermagnetic-ferrimagnetic transitions. Instead, it has been reported that a method of maintaining superparamagnetic behaviour with high magnetization by forming magnetic nanoclusters is effective. Magnetic nanoparticles composed of multiple single MNCs are particularly attractive due to their high magnetic susceptibility, low coercive force and high magnetic properties. 

References

1.    H. Lee, T.-J. Yoon, J.-L. Figueiredo, F. K. Swirski and R. Weissleder, Proc. Natl. Acad. Sci. U. S. A., 2009, 106, 12459-12464.

2.    Y. I. Park, Y. Piao, N. Lee, B. Yoo, B. H. Kim, S. H. Choi and T. Hyeon, J. Mater. Chem., 2011, 21, 11472-11477.

3.    U. I. Tromsdorf, N. C. Bigall, M. G. Kaul, O. T. Bruns, M. S. Nikolic, B. Mollwitz, R. A. Sperling, R. Reimer, H. Hohenberg, W. J. Parak, S. Förster, U. Beisiegel, G. Adam and H. Weller, Nano Lett., 2007, 7, 2422-2427.

4.   Y.-w. Jun, Y.-M. Huh, J.-s. Choi, J.-H. Lee, H.-T. Song, S. Yoon, K. –S. Kim, J. –S. Shin, J.-S. Suh and J. Cheon. J. Am. Chem. Soc., 2005, 127, 5732-5733.

5.   J.-H. Lee, Y.-M. Huh, Y. –w. Jun, J.-w. Seo, J.-t. Jang, H.-T. Song, S. Kim. E. –J. Cho, H.-G. Yoon, J.-S. Suh and J. Choen. Nat. Med., 2007, 13, 95-99.

6.    E. Pöselt, H. Kloust, U. Tromsdorf, M. Janschel, C. Hahn, C. Maßlo and H. Weller, ACS Nano, 2012, 6, 1619-1624.

7.    T.-J. Yoon, H. Lee, H. Shao, S. A. Hilderbrand and R. Weissleder, Adv. Mater., 2011, 23, 4793-4797.

8.    P. Qiu, C. Jensen, N. Charity, R. Towner and C. Mao, J. Am. Chem. Soc., 2010, 132, 17724-17732.

With reference to this prior literatures, we formulated magnetic nanoclusters using 12 nm MnFe2O4 nanocrystals for highly sensitive MRI contrast agents with excellent dispersibility.

Page 2, line 55: There is a limitation in increasing the size of MNCs to enhance saturation magnetization (Ms) because it also induces the transition between the supermagnetic-ferrimagnetic transitions. Instead, it has been reported that a method of maintaining superparamagnetic behaviour with high magnetization by forming magnetic nanoclusters (or assemblies) is effective. Magnetic nanoparticles composed of multiple single MNCs are particularly attractive due to their high magnetic susceptibility, low coercive force and high magnetic properties [44-46].

Page 5, line 118: Despite the 18.7 wt% mPEG-PLA coating, superparamagnetism of 12 nm-MNCs was maintained with a high saturated magnetization value (40.3 emu/gMNCs) at 298 K (Figure 5b) [44,45].

Page 6, line 132: These findings demonstrated that PEGylated MNs possessed a remarkably high MR imaging effect due to the enhanced magnetism through dense clustering of large amount of 12 nm MnFe2O4 (MNCs) in the PEGylated MNs [44-46].

6. Finally, there is a lot of information missing from the MRI part:

- How many subjects were analyzed?

: We performed MR imaging experiment using 5 mice. Related sentence has been added in the manuscript.

Page 9, line 239: Then, MR imaging was performed using 5 mice 4 weeks after tumor cell transplantation.

- The absence of error bars in Fig 8 suggests only one. If that is the case it is insufficient for a figure.

: We have changed to Figure 8 (c) containing error bars.  

Figure 8. (a) T2-weighted MR images of mice and (b) color-map images of polygonal region with white dashed line of (a). (c) ΔR2/R2Pre (%) graph of T2-weighted MR images versus the time after intravenous vein (I.V.) injection of PEGylated MNs (Pre: pre-injection, IMM: immediately following the injection, 1h: 1 h following the injection, 3 h: 3 h following the injection, and 7h: 7 h following the injection).

- Why were the other organs not analysed?

: It is well known the advantages of PEGylation on nanoparticle surfaces: 1) long-term circulation, 2) reduced accumulation of reticuloendothelial system (RES) related organs (liver and spleen) and 3) increased accumulation of tumor sites (H. Gao, J. Liu, C. Yang, T. Cheng, L. Chum H. Xu, A. Meng, S. Fan, L. Shi, J. Liu, Int. J. Nanomedicine., 2013, 8, 4229-42246). Of course, most of them accumulate in RES related organs, and we have confirmed this in previously our published papers (E.K. Lim, et. al. Adv. Mater. 2011, 23, 2436-2442; E.K. Lim, et. al.  Nanotechnology 2014, 25, 245103-245112; E.K. Lim, et. al.J Biomed Mater Res A 2014, 102, 49-59; E. K. Lim et. al. Biomaterials 2011, 32, 7941-7950; E.K. Lim et. al.Biomaterials 2010, 31, 9310-9319; E.K. Lim et. al. J. Mater. Chem. 2011, 21, 12473-12478). Hassan et. al. studied that the bio-distribution of PEG-PCL micelles in normal Balb/c mice over 48 h using in vivo fluorescence imaging (H. Asem, Y. Zhao, R. Ye, Å. Barrefelt, M. Abedi-Valugerdi, R. El-Sayed, I. Ei-Serafi et al., J. Nanobiotechnol. 2016, 14, 82-98). They confirmed that PEG-PCL micelles were highly distributed into the lungs during the first 4 h post intraveneous administration, then redistributed and accumulated in liver and spleen until 48 h post administration.  

 In this study, we developed magnetic nanoclusters in which highly sensitive magnetic nanocrystals were uniformly assembled using PEG-based amphiphilic polymer (mPEG-PLA), and then applied it to an in vivo model to focus on evaluating whether it is suitable as an MR imaging contrast agent for cancer diagnosis.

- How was the sample introduced, e.g. bolus tail-vein injection? If so at what volume, and at what concentration? What was the medium isotonic?

: 200 μg (Fe+MN concentration) of PEGylated MNS dispersed in a 200 μL buffer were injected into the tail vein of the mouse. Also, related content is added in detail in the manuscript.

Page 7, line 156: Next, we performed MR imaging in mouse xenograft tumor models using PEGylated MNs to evaluate their diagnostic ability as MR imaging contrast agents. First, BALB/c-nude mice were subcutaneously implanted in the proximal thigh with NIH3T6.7 cells for xenograft mouse model, and PEGylated MNs (200 μgFe+Mn in 200 μL buffer) were injected into the tail vein of the mouse (intravenous injection).

Reviewer 2 Report

The paper by Kang and colleagues, is well organized and the results a correctly presented. However, a detailed discussion on the “PEGylated magnetic nanoberries” advantages as contrast agents and comparison vs. other nanomaterials with similar properties is missing. This discussion would be beneficial for the readers and the impact of the manuscript.

Author Response

Reviewer 2

The paper by Kang and colleagues, is well organized and the results a correctly presented. However, a detailed discussion on the “PEGylated magnetic nanoberries” advantages as contrast agents and comparison vs. other nanomaterials with similar properties is missing. This discussion would be beneficial for the readers and the impact of the manuscript.

: Thank you for reviewer’s valuable feedback. It is well-known that magnetic relaxivity is influences by three factors: size and crystallinity of magnetic nanocrystals (MNCs), and magnetic nanoparticles’ size. First, the size of MNCs is directly dependent on the magnetic sensitivity (relaxivity, r2) which has been experimentally proven by many researchers (H. Lee, T.-J. Yoon, J.-L. Figueiredo, F. K. Swirski and R. Weissleder, Proc. Natl. Acad. Sci. U. S. A., 2009, 106, 12459-12464; Y. I. Park, Y. Piao, N. Lee, B. Yoo, B. H. Kim, S. H. Choi and T. Hyeon, J. Mater. Chem., 2011, 21, 11472-11477; U. I. Tromsdorf, N. C. Bigall, M. G. Kaul, O. T. Bruns, M. S. Nikolic, B. Mollwitz, R. A. Sperling, R. Reimer, H. Hohenberg, W. J. Parak, S. Förster, U. Beisiegel, G. Adam and H. Weller, Nano Lett., 2007, 7, 2422-2427).

 Cheon et. al. reported the effect of magnetic sensitivity according to the size and composition of MNCs (Y.-w. Jun, Y.-M. Huh, J.-s. Choi, J.-H. Lee, H.-T. Song, S. Yoon, K. –S. Kim, J. –S. Shin, J.-S. Suh and J. Cheon. J. Am. Chem. Soc., 2005, 127, 5732-5733; J.-H. Lee, Y.-M. Huh, Y. –w. Jun, J.-w. Seo, J.-t. Jang, H.-T. Song, S. Kim. E. –J. Cho, H.-G. Yoon, J.-S. Suh and J. Choen. Nat. Med., 2007, 13, 95-99). MnFe2O4 nanoparticle showed the strongest MR contrast effect, with high relaxivity value (Figure R1). Especially, 12-nm MnFe2O4 nanoparticles showed the highest mass magnetization value (Figure R2) (J.-H. Lee, Y.-M. Huh, Y. –w. Jun, J.-w. Seo, J.-t. Jang, H.-T. Song, S. Kim. E. –J. Cho, H.-G. Yoon, J.-S. Suh and J. Choen. Nat. Med., 2007, 13, 95-99).

Figure R1. Magnetism-engineered iron oxide nanoparticles and effects of their magnetic spin on MRI, which is reproduced from ref (Nat. Med., 2007, 13, 95-99).

Figure R2. Size-dependent MR contrast effect of MnFe2O4 and Fe3O4 nanoparticles, which is reproduced from ref (Nat. Med., 2007, 13, 95-99).

However, increasing the size of MNCs to enhance saturation magnetization (Ms) is limited because it also induces the transition between the supermagnetic-ferrimagnetic transitions. Instead, it has been reported that a method of maintaining superparamagnetic behaviour with high magnetization by forming magnetic nanoclusters is effective. Magnetic nanoparticles composed of multiple single MNCs are particularly attractive due to their high magnetic susceptibility, low coercive force and high magnetic properties. 

Based on this references, we have formulated PEGylated magnetic nanoassemblies (PEgylate MNs) by clustering using 12 nm MnFe2O4 nanocrystals with PEG-based amphiphilic polymers for highly sensitive MRI contrast agents with excellent dispersibility. PEG molecules on the surface of PEGylated MNs ca be hydrogen bonded to water molecules to avoid the clearance by macrophage cells of the reticuloendothelial system (RES), enabling long blood circulation time in the body, named PEGylation effect. As a results, PEGylation MNs can be delivered and accumulated to the desired sites (e.g. tumor tissue) by EPR effect, which contributes to accurate disease diagnosis using MR imaging. 

Based on the comments of the reviewer, we have revised and supplemented the contents as a whole.

Page 2, line 55: There is a limitation in increasing the size of MNCs to enhance saturation magnetization (Ms) because it also induces the transition between the supermagnetic-ferrimagnetic transitions. Instead, it has been reported that a method of maintaining superparamagnetic behavior with high magnetization by forming magnetic nanoclusters (or assemblies) is effective. Magnetic nanoparticles composed of multiple single MNCs are particularly attractive due to their high magnetic susceptibility, low coercive force and high magnetic properties [44-46].

Page 5, line 118: Despite the 18.7 wt% mPEG-PLA coating, superparamagnetism of 12 nm-MNCs was maintained with a high saturated magnetization value (40.3 emu/gMNCs) at 298 K (Figure 5b) [44,45].

Page 6, line 132: These findings demonstrated that PEGylated MNs possessed a remarkably high MR imaging effect due to the enhanced magnetism through dense clustering of large amount of 12 nm MnFe2O4 (MNCs) in the PEGylated MNs [44-46].

Page 7, line 156: Next, we performed MR imaging in mouse xenograft tumor models using PEGylated MNs to evaluate their diagnostic ability as MR imaging contrast agents. First, BALB/c-nude mice were subcutaneously implanted in the proximal thigh with NIH3T6.7 cells for xenograft mouse model, and PEGylated MNs (200 μgFe+Mn in 200 μL buffer) were injected into the tail vein of the mouse (intravenous injection).

Page 8, line 165: The color map image is intended to make the R2 value (MR signal) change value visually clear. In a color map image, as MR signal increase, the color changes from blue to red. Therefore, in Figure 8b, the red color spread out gradually along the vascular distribution as the MR signal increased, similar to the T2-weighted MR images.

Page 8, line 196: After the reaction was terminated, the heat source was removed from reactant and their temperature was cooled down to room temperature. The solvent of this reactant was rapidly eliminated using rotary evaporator (50 HZ, EYELA). For purification, this reactant was re-dissolved in toluene (2 mL) and precipitated in cold diethyl ether, and then it was filtered using a vacuum filtration. This purification process was repeated three times. The reactant was freeze-dried to obtain purified product as a white powder and stored under a vacuum before use.

Page 9, line 239: Then, MR imaging was performed using 5 mice 4 weeks after tumor cell transplantation.

Page 9, line 262: At this time, the MNCs were uniformly aggregated in raspberry form, and we named it PEGylated magnetic nano-assemblies (PEGylated MNs).

In addition, we changed to Figure 8(c).

Figure 8. (a) T2-weighted MR images of mice and (b) color-map images of polygonal region with white dashed line of (a). (c) ΔR2/R2Pre (%) graph of T2-weighted MR images versus the time after intravenous vein (I.V.) injection of PEGylated MNs (Pre: pre-injection, IMM: immediately following the injection, 1h: 1 h following the injection, 3 h: 3 h following the injection, and 7h: 7 h following the injection).

Reviewer 3 Report

This manuscript deals with the preparation, characterization and MRI studies of nanoparticles coated with a polymer. The authors show that they form stable NP, with low toxicity and generating T2 contrast mainly through EPR effect.

My main concern with this paper is the absence of comparison with literature data. There are numerous reports on NP generating T2 contrast, so what are the advantages/disadvantages of this system compared to the literature in terms of design, efficacy, specificity etc… This should be discussed honestly.

I have also concerns with the in vivo MRI experiments, how many mice were studied? What are the errors on the values obtained? The contrast is similar at 2 and 3h and then increase for 7h. Is this significant? If yes, what does it mean? Is it a normal behavior? Did you study the clearance of the contrast agent?

Here are also a few minor remarks:

-          Please define MNC the first time you use it in the text.

-          Please indicate the temperature and field at which you measure r2 in figure 6 and the corresponding text. Please change R2 by r2 in figure 6. R2 is the relaxation rate (1/T2 in s-1), while r2 is the relaxivity in mM-1.s-1. These values should not be mixed.

-          Please define the field of the MRI experiments within the main text (discussion) and in Figure 8. Please define also the type of mouse you are injecting in the results and discussion, and not only in the materials and methods.

-          In figure 8b, please define also anatomically the area you are looking at (not just rectangular region)

-          Line 157, the sentence: “In Figure 8b, the red color spread out gradually along the vascular

distribution as the MR signal increased, similar to the T2‐weighted MR images » is misleading as Figure 8b represent T2-weighted images, just in a color version.

Author Response

Reviewer 3

This manuscript deals with the preparation, characterization and MRI studies of nanoparticles coated with a polymer. The authors show that they form stable NP, with low toxicity and generating T2 contrast mainly through EPR effect. My main concern with this paper is the absence of comparison with literature data. There are numerous reports on NP generating T2 contrast, so what are the advantages/disadvantages of this system compared to the literature in terms of design, efficacy, specificity etc… This should be discussed honestly.

: Thank you for reviewer’s valuable feedback. In MR imaging, the contrast enhancement effects are directly related to the Ms value of the nanoparticles. Specifically, transverse relaxivity (r2) represents the degree of T2-weighted MRI contrast effect where the r2 value is proportional to the Ms value. As shown in Table R1, we compared MR contrast effect with our agent (PEGylated MNs) and several iron-based MR contrast agents (Ferumoxide, cross-linked iron oxide(CLIO).

Contrast agent

Magnetic core

Core diameter (nm)

Surface coating material
  (Surfactant)

Hydrodynamic size (nm)

r2 (mM-1s-1)

PEGylated MNs

MnFe2O4

12

mPEG-PLA

65-70

217.1

CLIO

Fe3O4

5

Dextran

30

62

Ferumoxide

(Feridex I.V.TM)

Fe3O4

γ-Fe2O3

4.96

Dextran

80-150

190.5

 Besides, it was compared the physicochemical characterizations of some commercial MRI contrast agents and novel superparamagnetic iron oxide nanoparticles (SPION) in the literature (L. Li, W. Jiang, K. Luo, H. Song, F. Lan, Y. wu and Z. Gu, Theranostics, 2013, 3, 595-615). Among them, our PEGylated MNs were confirmed to be of sufficient value as MRI contrast agent.

I have also concerns with the in vivo MRI experiments, how many mice were studied? What are the errors on the values obtained? The contrast is similar at 2 and 3h and then increase for 7h. Is this significant? If yes, what does it mean? Is it a normal behavior? Did you study the clearance of the contrast agent?

: We performed MR imaging experiment using 5 mice and have revised related sentence and Figure 8 (c) containing error bars in the manuscript. And, as shown in revised Figure 8 (c), the MR signal at 3 hours was slightly increased compared to 2 hours. This means that PEGylated MNs continue to circulated in the body and are delivered to the tumor. We are sorry to have not confirmed the clearance of PEGylated MNs in the body.

Figure 8. (a) T2-weighted MR images of mice and (b) color-map images of polygonal region with white dashed line of (a). (c) ΔR2/R2Pre (%) graph of T2-weighted MR images versus the time after intravenous vein (I.V.) injection of PEGylated MNs (Pre: pre-injection, IMM: immediately following the injection, 1h: 1 h following the injection, 3 h: 3 h following the injection, and 7h: 7 h following the injection).

 Here are also a few minor remarks:

-          Please define MNC the first time you use it in the text.

-          Please indicate the temperature and field at which you measure r2 in figure 6 and the corresponding text. Please change R2 by r2 in figure 6. R2 is the relaxation rate (1/T2 in s-1), while r2 is the relaxivity in mM-1.s-1. These values should not be mixed.

: We have corrected this error.

-          Please define the field of the MRI experiments within the main text (discussion) and in Figure 8. Please define also the type of mouse you are injecting in the results and discussion, and not only in the materials and methods.

: We have corrected this point in the experimental section.

Page 7, line 156: Next, we performed MR imaging in mouse xenograft tumor models using PEGylated MNs to evaluate their diagnostic ability as MR imaging contrast agents. First, BALB/c-nude mice were subcutaneously implanted in the proximal thigh with NIH3T6.7 cells for xenograft mouse model, and PEGylated MNs (200 μgFe+Mn in 200 μL buffer) were injected into the tail vein of the mouse (intravenous injection).

-          In figure 8b, please define also anatomically the area you are looking at (not just rectangular region)

: We have modified corresponding parts as follows.

Figure 8. (a) T2-weighted MR images of mice and (b) color-map images of polygonal region with white dashed line of (a). (c) ΔR2/R2Pre (%) graph of T2-weighted MR images versus the time after intravenous vein (I.V.) injection of PEGylated MNs (Pre: pre-injection, IMM: immediately following the injection, 1h: 1 h following the injection, 3 h: 3 h following the injection, and 7h: 7 h following the injection).

-          Line 157, the sentence: “In Figure 8b, the red color spread out gradually along the vascular distribution as the MR signal increased, similar to the T2weighted MR images » is misleading as Figure 8b represent T2-weighted images, just in a color version.

: The relevant sentence was modified as follows.  

Page 8, line 165: The color map image is intended to make the R2 value (MR signal) change value visually clear. In a color map image, as MR signal increase, the color changes from blue to red. Therefore, in Figure 8b, the red color spread out gradually along the vascular distribution as the MR signal increased, similar to the T2-weighted MR images.

Round  2

Reviewer 1 Report

The first revision of the manuscript “PEGylated magnetic nanoberries as T2-weighted MR contrast agents….” by Lim and co-workers addresses all but two of the questions raised.

1. “The DLS analysis is incomplete; the uncertainties given are presumably from repeat measurements. What was the PDI and how do the dhyd and PDI compare to the statistical TEM analysis? The Malvern instrument was not used, but some measure of polydispersity must be possible. The suspensions are stable in that the dhyd value didn’t change, but are the PDI and concentration unchanging (the DLS counts could help for the latter).

The value now obtained 0.378 indicates a suspension of marginal quality, however, it is not included ini the revised manuscript. Were the Malvern instrument used it would be deemed outside the normal range. I suggest that the DLS value is discussed briefly in the doc and the corallelograms, including the cumulants fit that was used to generate the PDI value be included as a supplementary figure.

If the quality of this analysis proves to be marginal, a DLS demonstration of colloidal stability e.g. DLS size distributions or corallelograms as a function of time might suffice; again a fig for SI should be provided.

2. “My major criticism is the absence of a comparative analysis of the relaxivity versus the literature; there are a great many papers including a number on polymer stabilised clusters of this type. I know some of this work and r2=217s-1mM-1 (1.5T) at c.65 nm seems reasonable if not very high, but it is for the Authors to complete this analysis. There are at least three points; the MNCs size and crystallinity; the cluster size, and; the polymer (weight%, mol wt and relative wts of the two blocks).

This has now been provided in the response letter, but it remains completely absent from the manuscript. A comparative analysis (perhaps shorter than was provided but including discussion of the r2 values particularly as compared to clusters of comparable =size) should be included in the paper.

Finally, the authors have retained the phrase “magnetic sensitivity" in the abstract. I don’t understand what this means.

Author Response

Reviewer 1

The first revision of the manuscript “PEGylated magnetic nanoberries as T2-weighted MR contrast agents….” by Lim and co-workers addresses all but two of the questions raised.

1. “The DLS analysis is incomplete; the uncertainties given are presumably from repeat measurements. What was the PDI and how do the dhyd and PDI compare to the statistical TEM analysis? The Malvern instrument was not used, but some measure of polydispersity must be possible. The suspensions are stable in that the dhyd value didn’t change, but are the PDI and concentration unchanging (the DLS counts could help for the latter).

The value now obtained 0.378 indicates a suspension of marginal quality, however, it is not included ini the revised manuscript. Were the Malvern instrument used it would be deemed outside the normal range. I suggest that the DLS value is discussed briefly in the doc and the corallelograms, including the cumulants fit that was used to generate the PDI value be included as a supplementary figure.

If the quality of this analysis proves to be marginal, a DLS demonstration of colloidal stability e.g. DLS size distributions or corallelograms as a function of time might suffice; again a fig for SI should be provided.

: We thank valuable reviewer’s comments. Based on your advice, we have added relevant sentences to this manuscript.

Page 4, line 113:

As well, the PDI values were calculated based on the DLS analysis results. When each value was substituted into the PDI equation (=Standard deviation2/ Size), the PDI value were 0.38, 0.06 and 0.49, respectively (Table 1 and Figure S1). The PDI values of the particles ranged from 0 to 0.0.8 in nearly monodisperse and 0.08 to 0.7 in uniformly dispersed [52]. Based on our PDI values, we judged that PEGylated MN exhibited uniformly disperse and acceptable to use in the pharmaceutical filed.

Table 1. The size, PDI values and zeta potential data of PEGylated MN over 44days.

Time

Size   (nm)

PDIa

Zeta   (mV)

0   day

68.8 ± 5.1

0.38

0.7 ± 0.3

33   day

69.2 ± 2.0

0.06

-2.4± 0.7

44 day

63.2 ± 5.6

0.49

-1.1± 0.9

All data are depicted as the mean ± S.D, and N > 3, aPDI = (S.D.)2/Avg. size

52.           Danaei, M.; Dehghankhold, M.; Atae, S.; Hasanzadeh Davarani, F.; Javanmard, R.; Dokhani, A.; Khorasani, S.; Mozafari, M. R. Impact of Particle Size and Polydispersity Index on the Clinical Applications of Lipidic Nanocarrier Systems. Pharmaceutics. 2018, 10, 57-74.

Also, all results of the size analysis were added in the Supporting materials (Figure S1).

Figure S1. The size analysis graphs of PEGylated MNs over 44 days (a): 0 day, b): 30 day and c): 44day). All measurements were repeated three times.

2. “My major criticism is the absence of a comparative analysis of the relaxivity versus the literature; there are a great many papers including a number on polymer stabilised clusters of this type. I know some of this work and r2=217s-1mM-1 (1.5T) at c.65 nm seems reasonable if not very high, but it is for the Authors to complete this analysis. There are at least three points; the MNCs size and crystallinity; the cluster size, and; the polymer (weight%, mol wt and relative wts of the two blocks).

This has now been provided in the response letter, but it remains completely absent from the manuscript. A comparative analysis (perhaps shorter than was provided but including discussion of the r2 values particularly as compared to clusters of comparable =size) should be included in the paper.

 : We added related sentences in the manuscript, which was shown below.

Page 5, line 127:

It was confirmed that the magnetic sensitivity of MNCs was affected by their size and composition [44,45]. Of the 4-types MNCs (MnFe2O4, Fe3O4, CoFe2O4 and NiFe2O4), MnFe2O4 nanocrystals showed the strongest MR contrast effect, with high relaxivity values. Especially, 12-nm MnFe2O4 nanocrystals exhibited the highest mass magnetization value. Based on this previous references, we used 12-nm MnFe2O4 nanocrystals as MNCs to fabricate PEGylated MNs.

Finally, the authors have retained the phrase “magnetic sensitivity" in the abstract. I don’t understand what this means.

: We also changed the word mentioned by the reviewer (magnetic sensitivity) to the “enough MR contrast effect”.

 I hope I answered your question completely.

Reviewer 3 Report

The authors have revised their manuscript taking my remarks into account only partially. For example, I would like to see in the main text and in Figure 6 what are the field and temperature at which r2 was determined. This is important, also when comparing to other values. I would also like to see the magnetic field of the in vivo MRI experiment in Figure 8. This has not been corrected in the last version of the manuscript. I do not see either in the legend of figure 8 what is the anatomical part of the mouse which was studied.

Finally I regret that their system is only superficially compared to litterature data in terms of relaxivity. A sentence on the originality, advantages/disadvantages of such design compared to what is done in the litterature would have really be welcomed.

Author Response

Reviewer 3

The authors have revised their manuscript taking my remarks into account only partially. For example, I would like to see in the main text and in Figure 6 what are the field and temperature at which r2 was determined. This is important, also when comparing to other values. I would also like to see the magnetic field of the in vivo MRI experiment in Figure 8. This has not been corrected in the last version of the manuscript. I do not see either in the legend of figure 8 what is the anatomical part of the mouse which was studied.

: First, we measured MR signal intensities in aqueous phase at room temperature and added this experimental temperature to the manuscript as shown below to aid understanding.

Page 6, line 140:

We evaluated the feasibility of PEGylated MNs as MR imaging agents and measured their MR signal intensities in an aqueous phase under various concentrations at room temperature.

The second, we have added an anatomical part of the mouse in Figure 8, and corrected the critical error additionally. 

Also, Figure 6 and 8’s captions were revised.

Figure 6. T2-weighted MR images of PEGylated MN solution and their color maps, and their 1/T2 (S-1) values at 1.5 T.

Figure 8. (a) T2-weighted MR images of mice and (b) color-map images of polygonal region with white dashed line of (a). (c) ΔR2/R2Pre (%) graph of T2-weighted MR images versus the time after intravenous vein (I.V.) injection of PEGylated MNs (Pre: pre-injection, IMM: immediately following the injection, 1h: 1 h following the injection, 3 h: 3 h following the injection, and 7h: 7 h following the injection). A 3.0 T human MR scanner was used.

Finally I regret that their system is only superficially compared to litterature data in terms of relaxivity. A sentence on the originality, advantages/disadvantages of such design compared to what is done in the litterature would have really be welcomed. 

:  The r2 of the ferumoxide (Feridex I.V.TM) (190.5 mM-1s-1) mentioned in the manuscript was measured with the same equipment and condition to compared with the r2 value of PEGylated MNs. Therefore, we judged that PEGylated MN is worth using as MRI contrast agent by comparing MR sensitivity (r2 value) with commercial MRI contrast agent (ferumoxide).

 We have modified the conclusion section (Page 10).

In this study, we synthesized mPEG-PLA as an amphiphilic polymer using the ring-opening polymerization method and then encapsulated the hydrophobic magnetic nanocrystals (MNCs) in organic solvent with mPEG-PLA using the nano-emulsion method to allow stable dispersion in the aqueous phase. At this time, the MNCs were uniformly assembled in raspberry form, and we named it PEGylated magnetic nano-assemblies (PEGylated MNs). The PEGylated MNs exhibited good stability in an aqueous phase for an extended time due to the PEG molecules on the particle surface (PEGylation effect) as well as an enough MR contrast effect due to the magnetic clustering effect compared to those of a commercial MR contrast agent. In addition, we confirmed that PEGylated MNs had potential use as MRI agents for cancer detection through in vivo studies.

Round  3

Reviewer 1 Report

The second revision of the manuscript “PEGylated magnetic nanoberries as T2-weighted MR contrast agents….” by Lim and co-workers is acceptable.